# Adsorption of Cd to TiO_2_-NPs Forms Low Genotoxic Aggregates in Zebrafish Cells

**DOI:** 10.3390/cells10020310

**Published:** 2021-02-03

**Authors:** Filomena Mottola, Marianna Santonastaso, Concetta Iovine, Veronica Feola, Severina Pacifico, Lucia Rocco

**Affiliations:** 1Department of Environmental, Biological and Pharmaceutical Sciences and Technologies, University of Campania “Luigi Vanvitelli”, 81100 Caserta, Italy; filomena.mottola@unicampania.it (F.M.); concetta.iovine@unicampania.it (C.I.); veronica.feola@studenti.unicampania.it (V.F.); severina.pacifico@unicampania.it (S.P.); 2Department of Woman, Child and General and Special Surgery, University of Campania “Luigi Vanvitelli”, 80138 Napoli, Italy; marianna.santonastaso@unicampania.it

**Keywords:** DNA damage, apoptotic cells, genome instability, titanium dioxide nanoparticles, cadmium, zebrafish

## Abstract

The aquatic environment is involved in the pollutants spreading mechanisms, including nanomaterials and heavy metals. The aims of this study were to assess the in vivo genotoxicity of Cd (1 mg/L) and to investigate the genomic effects generated by its co-exposure with TiO_2_-NPs (10 µg/L). The study was performed using zebrafish as a model for 5, 7, 14, 21, and 28 days of exposure. The genotoxic potential was assessed by three experimental approaches: DNA integrity, degree of apoptosis, and molecular alterations at the genomic level by genomic template stability (% GTS) calculation. Results showed an increased in DNA damage after Cd exposure with a decrease in % GTS. The co-exposure (TiO_2_-NPs + Cd) induced a no statistically significant loss of DNA integrity, a reduction of the apoptotic cell percentage and the recovery of genome stability for prolonged exposure days. Characterization and analytical determinations data showed Cd adsorption to TiO_2_-NPs, which reduced free TiO_2_-NPs levels. The results of our study suggest that TiO_2_-NPs could be used for the development of controlled heavy metal bioremediation systems.

## 1. Introduction

Titanium dioxide (TiO_2_) nanoparticles (NPs), continuously released into waters due to their widespread use, may cause harmful effects to aquatic organisms and their potential interaction with conventional toxic contaminants represents a growing concern for biota. 

Bioavailability and genotoxicity of TiO_2_-NPs for aquatic biota is well documented [1,2,3]. TiO_2_ is a mineral oxide, normally found in igneous rocks and widely used in the cosmetic, pharmaceutical, and paint industries [4]. It is considered physiologically inert and presents little risk for humans [5], however, when TiO_2_ is used at the nanometric scale, its biological and environmental effects can be different [6]. Our previous study showed that TiO_2_-NPs induces zebrafish (*Danio rerio*) DNA damage at concentrations very close to those found in the environment (1 and 10 μg/L). We showed that the induction of greater damage occurred at the highest concentrations tested and only for intermediate treatment times (14 and 21 days), suggesting the DNA defense activation and DNA repair mechanisms for prolonged exposure times [7]. These results are confirmed by numerous studies conducted on the zebrafish that was considered an easy model for toxicity tests and for the aquatic environments biomonitoring [8,9]. 

Antioxidant enzymes’ gene expression was upregulated to resist the adverse effects of TiO_2_-NPs, which is consistent with the basic defense mechanism of organisms [10]. However, an excessive amount of TiO_2_-NPs results in an over production of reactive oxygen species (ROS) with oxidative damage in various aquatic models both in vivo and in vitro. It has been observed that nanoparticles internalize cells causing an increase in intracellular ROS production with cytotoxicity and genotoxicity as a consequence of a reduced enzymatic markers of oxidative stress activity (superoxide dismutase (SOD), catalase (CAT), and glutathione (GSH) levels) [11]. Nevertheless, even at low doses, TiO_2_-NPs in anatase form cause cell damage despite the increase in antioxidant defenses in *Mytilus galloprovincialis* [12].

Besides the potential of TiO_2_-NPs to affect aquatic organisms directly through its inherent properties, it is also expected to interact with xenobiotics [13] and heavy metals modifying their availability and/or toxicity [14]. 

Heavy metals toxicity is a considerable problem for ecological, evolutionary, nutritional, and environmental reasons. Cadmium (Cd) has an increased diffusion because it is widely used in many industrial productions little or not controlled [15]. When Cd enters an organism, it can bioaccumulate in various tissues and biomagnifies through the food chain. Many aquatic species of economic food interest such as *Engraulis*, squid, bowfin, bivalves, have been found with high Cd levels compared to regulatory standards [16]. This condition is very dangerous as Cd is responsible for chromosomal alterations and DNA damage in that it leads to ROS production [16,17]. Moreover, Cd interferes with mitochondria by altering ROS balance [18,19] and/or inducing a lowering of various enzymes and antioxidants levels [20,21]. The DNA vulnerability to Cd and the different pathways that cells play out as a response, i.e., damage repair or cell death, have a key role in the onset and progression of cancer [22]. Maintaining the integrity and stability of DNA molecular structure is of great importance for cell survival and normal physiological function. In fact, a genotoxic insult can induce DNA strand breaks (DSB) and a loss of function in DNA damage response (DDR) mechanisms with consequent mutagenesis and transformation of the healthy cells into malignant ones [23].

Cd combination with widespread genotoxic molecules in the environment represents a danger for the biota with an increased risk for carcinogenesis due to a possible synergistic and/or antagonistic effect. In particular, the greatest risk is represented by Cd combination with NPs, known to be able to bind and transport other molecules, evading cellular defense mechanisms through the “Trojan horse effect” [24]. Literature data about effects due to TiO_2_-NPs and Cd interaction are conflicting. In fact, some evidence suggests that the interactions of TiO_2_-NPs with Cd lead to increased adverse effects in different biological systems and depend on the organisms and Cd dose [25,26]. Zhang and collaborators observed that Cd accumulation in carp tissues co-exposed to TiO_2_-NPs is increased by 146% compared to controls, due to the metal adsorption phenomena on the nanoparticulate material [27]. This interactions type is little explored, although aquatic organisms are often exposed to complex mixtures of contaminants, particularly when they live next to petrochemical poles, port, and/or industrial areas.

Some indications suggest that Cd and TiO_2_-NPs co-exposure prevents DNA damage with increased genomic stability in *Mytilus galloprovincialis* and in European sea bass. The combination of TiO_2_-NPs and Cd leads to formation of a less reactive compound, probably unable to penetrate cells, with consequent reduced genotoxicity [2].

This study aims to assess in vivo genotoxicity of Cd (1 mg/L) and to investigate the genomic effects generated by Cd co-exposure with TiO_2_-NPs (10 µg/L). The study was performed using zebrafish as a model for 5, 7, 14, 21, and 28 days of exposure. The genotoxic potential was assessed evaluating DSB by comet assay, degree of apoptosis by diffusion assay, and genomic alterations using RAPD-PCR technique; the last was used to calculate the genomic stability of the template (GTS %). These analyses in aquatic organisms have proved to be an effective method for the evaluation of genotoxic contamination in the environment and they will provide new data for understanding the interaction mechanisms between nanoparticles and environmental pollutants. 

## 2. Materials and Methods

### 2.1. Chemicals

TiO_2_-NPs (Aeroxide; Evonik Degussa, Essen, Germany; Lot. 614061098) stock suspensions (10 mg mL^−1^) was sonicated for 3 h at a frequency of 40 kHz, Dr. Hielscher UP 200S, Germany. UV Shimadzu 1700 Double Beam Spectrophotometer was used to obtain UV-Vis spectra in the range 200–600 nm [7]. TiO_2_-NPs solution was diluted to obtain end-point concentrations of 1 mg/L TiO_2_-NPs for exposure. Cadmium (CAS number 10108-64-2, 99.9% purity) was provided by Merck KGaA (Darmstadt, Germania). Benzene (CAS number 319953, 99.9% purity) was provided by Merck KGaA (Darmstadt, Germania).

### 2.2. Specimens Preparation

Experiments were conducted on 150 adult 8-month-old zebrafish purchased from a local supplier. Animal maintenance and handling and experimental procedures were in accordance with the Guide for Use and Care of Laboratory Animals (European Communities Council Directive) and efforts were made to minimize animal suffering and reduce the number of specimens used. 

After 2 weeks of acclimation, groups of 30 zebrafish were transferred to smaller tanks each containing 5 L of water in which the substances were dissolved at the chosen concentrations for 5, 7, 14, 21, and 28 days. We exposed zebrafish to Cd at concentration 1 mg/L, to 10 µg/L TiO_2_-NPs and to combination of Cd (1 mg/L) and TiO_2_-NPs (10 µg/L). 

Untreated specimens were used as negative controls, while 10 μL/mL benzene exposure was used as positive control for all tests. The water’s tanks were replaced and the substances were dissolved at the chosen concentrations every 7 days. The exposure scheme was the same for all the methods.

### 2.3. Specimens Sacrifice

The fish were sacrificed to collect blood and muscle for each experiment. In details, the zebrafish were anesthetized with Tricaine methalsulfonate (Sigma-Aldrich) according to the Guide for Use and Care of Laboratory Animals (European Communities Council Directive). The blood was collected by a heparinized syringe.

### 2.4. Characterization and Analytical Determinations 

TiO_2_-NPs and Cd content was spectrophotometrically determined by the UV-Vis spectroscopy according to Rocco and collaborators [7] with a method improvement which takes into account cadmium absorption. Free Cd was used as standard and its UV-Vis spectrum was recorded at 0.1, 1, 10, 25, 50, and 100 mg/mL in distilled water, and in tank water. The calibration curve (y = 0.0329x − 0.0016; R^2^ = 0.996) constructed based on Cd absorptions in tank water was used for quantitation purposes. Moreover, the TiO_2_-NPs dose level was estimated based on TiO_2_-NPs calibration curve in co-treated samples.

### 2.5. Comet Assay 

The comet assay (single cell gel electrophoresis) was performed according to Rocco and collaborators [7]. This test set up for the detection of DNA damage involves the encapsulation of blood cells in a suspension of low melting point agarose (0.7%) insert on the slide between two layers of normal melting agarose (1%); lysing cells under alkaline condition (pH > 12.1) to evidence also double strand breaks; and electrophoretic run at 25 V, 300 mA. The methodology requires the removal of few blood microliters from zebrafish gill because it is the organ directly exposed to water contaminants during the respiration [28]. Each slide was 1X ethidium bromide stained. 

We observed the slides using an epifluorescence microscope (Nikon Eclipse E-600) with a BP 515–560 nm filter and an LP 580 nm filter. The images were analyzed using an image acquisition and analysis software (Komet ver. 6.0.0, Kinetic Imaging), which allows to quantify the percentage of DNA in the comet’s tail. Comets were selected without bias and represented the entire gel as recommended by Collins [29]. The images from the comet slides were analyzed by the software by drawing a measurement frame on the screen around the comet area. The selected measurement parameters were saved and the statistical analysis was conducted. We considered relative fluorescence intensity of head and tail parameter (normally expressed as a percentage of DNA in the tail). The assay was performed in triplicate, three slides for sample were used for each experiment, and 50 cells were analyzed for each slide.

### 2.6. Diffusion Assay

The diffusion assay allows evaluating the percentage of apoptosis in isolated cells [30]. The test involves mixing the blood cells with low melting point agarose (0.7%) and the preparation of a normal melting agarose (1%) micro gel on a glass slide, the cells lysis with salts and detergents, and alkaline treatment according to Rocco and collaborators [7]. The method is similar to the comet assay, with the exception of the electrophoresis step [31]. The apoptotic cells show irregular edges with nuclei characterized by a highly dispersed DNA. The nuclei of necrotic cells are larger and not well defined [30]. We observed the slides using an epifluorescence microscope (Nikon Eclipse E-600) with a BP 515-560 nm filter and an LP 580 nm filter. 

The diffusion assay slides were scored by subdividing the degree of DNA diffusion pattern in five damage classes as reported by Cantafora and collaborators [32]. In details, we considered only class 5 (apoptotic cell). The number of apoptotic cells were quantified as the percentage of cells with apoptotic appearance on total cells by using a fluorescent microscope. The assay was performed in triplicate, three slides for sample were used for each experiment, and 50 cells were analyzed for each slide.

### 2.7. RAPD-PCR Technique and %GTS Calculation 

The RAPDs protocol provided for DNA isolation from zebrafish muscle samples using a commercial kit (High Pure PCR Template Preparation Kit, ROCHE Diagnostics). The DNA purity was evaluated by Nanodrop 2000 (Thermo Scientific), while DNA integrity thanks to electrophoresis on 2% agarose gel. The samples were analyzed by RAPD-PCR immediately after DNA isolation. We carried out amplification of the genomic DNA using the Primer 6 (5′-CCCGTCAGCA-3′) as we have already tested the DNA fingerprinting generated by Primer 6 in our previous studies [33]. The DNA amplification of each sample was obtained by PCR using the following cyclic amplification program: 2 min at 94 °C (denaturation of double-stranded DNA), 1 min at 95 °C, 1 min at 36 °C (annealing of primers), 2 min at 72 °C (extension of the amplification) for 44 cycles. The reaction products were analyzed by means of electrophoresis on 2% agarose gel, staining with 1× ethidium bromide. 

The polymorphic pattern generated by the RAPD-PCR profiles allowed the calculation of the genomic template stability percentage (GTS, %) as follows: GTS = (1 − *a*/*n*) × 100,(1)
where *a* is the average number of polymorphic bands detected in each exposed sample and *n* the number of total bands in the non-treated cells. Polymorphism in RAPD profiles considered the bands that appear and disappear compared to the control [34]. The changes in band are linked to molecular events. In particular, the appearance of bands identify that DNA damage can be due to point mutations and/or DNA rearrangements; while the disappearance of bands could be due to the formation of DNA adducts or double strand breaks also [35]. This statistical analysis allowed us to understand the variation in genomic stability following exposure to Cd and the co-exposure to Cd + TiO_2_-NPs. We considered changes in these values as a percentage of their negative controls (set to 100%). The assay was performed in triplicate. We isolated DNA from each treated zebrafish and made three DNA pools for each treatment. Then, each DNA pool underwent RAPD amplification and electrophoresis.

### 2.8. Statistical Analysis

The experimental data were expressed as mean ± standard deviation (SD). Differences in the percentages of DNA integrity, apoptotic cells, and GTS among the experimental groups were compared using ANOVA (analysis of variances) test by GraphPad Prism 6. The effect was considered significant when *p*-value ≤ 0.05.

## 3. Results

### 3.1. Characterization and Analytical Determinations

As the determination of TiO_2_-NPs by UV-Vis spectroscopy was fairly accurate, free Cd was used as standard in distilled water and its UV-Vis spectrum was recorded at different concentrations. Two weak absorption bands were detectable as shoulders at 228 and 244 nm. When Cd-treated water was investigated, an upshift peak at 265 nm was found, probably due to the Cd tendency to form stable soluble complexes in water. Thus, in order to quantize Cd in water, without distinguishing its complexity or identity, a new calibration curve was prepared by Cd freshly added to the tank water. Based on the absorbance values at 265 nm, the calibration curve was found with an equation equal to y = 0.0329x − 0.0016. Collected data were interpolated, and Cd concentration was measured to be 0.765 mg/L. Cd+TiO_2_-NPs sample analysis highlighted, beyond a maximum peak at 225 nm, a band at 346 nm, which was in line with an upshift (+24 nm) of the characteristic peak for dispersed TiO_2_-NPs nanoparticles [7]. This could be due to Cd adsorption to TiO_2_-NPs, which impoverished free TiO_2_-NPs levels. In fact, low TiO_2_-NPs concentration (1.72 μg/L) was recorded with respect to its nominal added concentration (Figure 1). 

### 3.2. Comet Assay 

The values obtained by comet assay for the zebrafish exposed to Cd (1 mg/L) indicate a statistically significant percentage (%) of DNA fragmentation for all treatment time with values ranging from 50.91 ± 0.81 (5 days) to 53.83 ± 0.37 (28 days). 

TiO2-NPs (10 μg/L) exposure induced a percentage of DNA fragmentation ranging from 19.28 ± 0.27 after 5 days to 31.14 ± 0.50 after 28 days; in particular, the data indicated that TiO2-NPs induced a statistically significant DNA damage only for intermediate exposure times with a percentage of DNA fragmentation of 30.40 ± 1.05 and 36.61 ± 0.99 after 14 and 21 days, respectively. 

Whereas, the combination of Cd (1 mg/L) and TiO2-NPs (10 μg/L) induced DNA fragmentation reduction with respect to the singles exposure, with values ranging from 47.09 ± 0.26 (5 days) to 29.87 ± 0.45 (28 days). In details, the results showed that starting from 14 treatment days to 28 exposure day, the % DNA fragmentation values were no longer significative with respect to the control, with percentage of 25.45 ± 0.63 after 14 exposure days and 30.01 ± 1.43 and 29.87 ± 0.45 after 21 and 28 days, respectively (Figure 2 and Figure 3). 

### 3.3. Diffusion Assay

DNA damage was detected considering the percentage of apoptotic cells recognized by means of the typical halo of DNA diffusion. 

Cd exposure induced a statically significant average of apoptotic cells increased starting from 5 exposure day and for all time, with values ranging from 10.74 ± 0.40 (5 days) to 15.33 ± 0.65 (28 days). 

TiO2-NPs (10 μg/L) induced percentage of apoptotic cells ranging from 6.92 ± 0.40 after 5 treatment days to 8.79 ± 0.56 after 28 days. The percentage of statistically significant diffusion data was found only after 14 and 21 days of exposure to TiO2-NPs with values equal to 9.91 ± 0.14 (14 days) and 13.03 ± 0.15 (21 days).

The Cd and TiO2-NPs co-exposure produced apoptosis in zebrafish only after early treatment co-exposure days, with 10.14 ± 0.53 and 8.81 ± 0.17 in percentage of apoptotic cells for 5 and 7 days, respectively. Finally, no statistically significant apoptotic process from intermediate to longer treatment time was observed, with values ranging from 5.74 ± 0.37 after 14 days to 5.67 ± 0.44 after 28 days (Figure 4 and Figure 5). 

### 3.4. RAPD-PCR Technique 

The RAPD-PCR technique evidenced variations of bands number in the treated samples compared to the negative control (Table 1). Non-treated zebrafish presented bands from 200 to 800 base pairs (bp). The electrophoretic profiles of each treated samples showed a different pattern from the negative control after each exposure days.

The exposure to Cd induced six bands variation after 5 days, seven bands variations after 7 and 14 days, while four and eight bands variations after 21 and 28 days, respectively. TiO_2_-NPs induced a variation of three bands after 5, 7, and 28 exposure days while seven and five bands after 14 and 21 exposure days respectively. The electrophoretic profiles obtained after Cd + TiO_2_-NPs exposure showed four bands changes after 5 and 14 exposure days; five bands changes after 7 days; and one band change after 21 and 28 exposure days with respect to the negative control.

### 3.5. Genomic Template Stability (GTS, °/°)

The percentage of genome stability was calculated from the analysis of RAPD profiles (Figure 6). Cd induced a statistically reduction of zebrafish DNA stability after all treatment days, while TiO_2_-NPs induced genome instability at 14 and 21 treatment days. The results showed how genomic stability increases after to Cd + TiO_2_-NPs co-exposure respect to the single exposure, until it almost reaches the negative control for long exposure times (28 days). 

## 4. Discussion

A wide range of pollutants, such as heavy metals and NPs, increasingly contaminate wastewater, and in addition to affecting aquatic organisms, can reach the human body through the food chain, causing serious damage to human health. The toxicity of titanium dioxide nanoparticles and cadmium on the biota focuses more attention because both are often present in different natural, working, domestic, or industrial environments. They perform a direct action, without being processed and/or transformed, on different cellular districts and deserve our attention, as, even if only contained in traces, they have an unclear biological importance. 

The effects of TiO_2_-NPs and Cd exposure are due to direct and indirect genotoxic actions. These actions can be mediated by the ability to generate ROS inside the cell or, to inhibit specific antioxidant enzymes [36,37,38,39], hence, with different mechanisms, these substances produce oxidative damage. It is known that, both TiO_2_-NPs and Cd can cross cells and generate DNA damage. TEM analysis showed that TiO_2_-NPs form small agglomerates capable of penetrating human sperm cells and generating DNA fragmentation and genomic instability by intracellular ROS production [40]. Cadmium penetrates the cell using the voltage-dependent calcium channels or alternatively the channels associated with transmembrane receptors, causing lipid peroxidation [41,42] and inducing mechanisms underlying carcinogenesis, such as DNA strand breaks and inhibition of DNA repair processes. 

Nowadays, considering the negative effects provoked by TiO_2_-NPs and Cd on biological macromolecules, the greatest concern is a possible synergistic/antagonistic effect given by their interaction.

The physic-chemical characteristics of NPs allow them to absorb compounds and easily cross various biological barriers with mechanisms not yet clear. The spread of NPs in biological systems seems to be facilitated by caveol systems and by endocytosis with a probable involvement of a transport system mediated by the ABC transport proteins such as P-glycoprotein (P-gp) [43]. Literature data showed how the incorporation of a molecule into nanoparticles allows it protection from degradation and/or premature inactivation in the organism and from drug resistance processes. The interactions between NPs and other substances, including aquatic pollutants, are so complex and could modify the amount of accumulated contaminants, but also amplify or alleviate their toxicity [44].

In this study, we investigated the effects of TiO_2_-NPs and Cd on zebrafish, evaluating the induction of apoptosis and the genome integrity and stability as a result of exposure to Cd and to its combination with TiO_2_-NPs for 5, 7, 14, 21, and 28 days. We used the comet and diffusion assay and the RAPD PCR analysis, as these methods have the advantage to identify the genotoxic impact without having a detailed knowledge of the identity and chemical–physical properties of the contaminants [45]. As TiO_2_-NPs could facilitate the transport of some contaminants into internal tissues [46], we have evaluated the genotoxicity in different tissues (muscle and erythrocytes) in order to fully investigate the genotoxicity given by the co-exposure of TiO_2_-NPs and Cd, as we previously performed in the marine environment species [3].

Our results confirm the cadmium genotoxicity; it shows its genotoxic effect from the first days of exposure, with statistically significant damage in terms of genome integrity reduction (DNA double-strand breaks, DSBs) as well as in genomic instability, which persists for all treatment days, and a time dependent increase in apoptotic cells percentage. These results are in agreement with the literature as cadmium, by acting on the mitochondria, induces an increase in ROS production, with oxidation of macromolecules, DNA damage and alterations in the repair mechanisms with consequent apoptosis [47]. 

Although ROS play an essential role in the control of various cellular processes, their accumulation generated by genotoxic compounds is dangerous for aquatic organism health and development due to their direct impact on DNA [48,49].

Cd and TiO_2_-NPs co-exposure revealed that their action is mutually inhibited following their interaction for prolonged exposure times, resulting in a marked decrease in DNA damage and a reduction in the apoptotic process compared to individual exposures. The phenomenon known as the “Trojan horse effect” does not occur as Cd is stored by the TiO_2_-NPs, which inhibits its genotoxic potential.

These results have a very important translational impact as the DSBs are significantly reduced. DSBs are the most lethal of all DNA lesions because they are more difficult to repair, in fact, they involve two paths: the union of the classical non-homologous ends (c-NHEJ) and homologous recombination (HR); their inefficient repair, operated by the DDR genes, can result in cancer [23].

Further, the formation of a genotoxic substances complex, no longer able to determine DSB in the exposed organisms, could represent a very important starting point for limiting tumorigenesis in organisms exposed to a mix of pollutants. DNA DSBs trigger the activation of oncogenes, inactivate tumor suppressors, and influence chemosensitivity and tumor progression, overall causing organisms to be more susceptible to cancers and other diseases [50].

The reduction of DSBs following co-exposure to Cd and TiO_2_-NPs are confirmed in other studies conducted on different models, such as human sperm cells, mussel and sea bass models, where the cadmium toxicity is reduced by the presence of TiO_2_ nanoparticles [2,3,51].

Our characterization results showed that the two substances aggregate in an aquatic environment as Cd adsorbs into TiO_2_-NPs and probably forms a compound unable to penetrate the cells and to damage the DNA. Furthermore, because of the interaction, the concentration of free TiO_2_-NPs nanoparticles decreases also, reducing the genotoxic effects of the latter. This phenomenon is observed at 14 days of exposure, and explains why in the first exposure days the damage caused by co-exposure to TiO_2_-NPs and Cd, is still statistically significant. 

Evidence of complex structures generated by the combination of metals and other contaminants with TiO_2_ nanoparticles derive from several studies; coexistence of As(III) and Cd^2^+ lead to formation of a ternary surface complex due to their synergistic adsorption into TiO_2_-NPs with consequent reduction release of the two contaminants into the water [52]. While the single presence of Cd leads to the formation of a “sandwich structure” with TiO_2_-NPs, which completely masks the Cd [51]. 

Although it is necessary to consider that sometimes these single substances have severe biological effects on organisms, the results of our study may lay the foundations for the development of controlled bioremediation systems. Bioremediation is a sustainable strategy for eliminating heavy metals present in different environment districts, one of these strategies is the biosorption; it is a complex process, which uses adsorption and absorption mechanisms to removal of contaminants from different environments [53]. 

In this context, nanotechnology represents a valid ally for purification of wastewater contaminated with heavy metals as the high surface/volume ratio of nanomaterials provides them with a high adsorption capacity. In fact, the exploitation of magnetic nanoparticles as adsorbents of contaminants in water treatment processes has already been in use for several years [26,54,55]. Although some adsorbents have been shown to be highly selective for Cd, such as silicate-titanate nanotubes embedded in a biodegradable hydrogel polymer [56], the application of TiO_2_-NPs bioremediation technology for Cd contamination is not yet carried out. 

Furthermore, the studies to date show the ability of the nanomaterials to reduce the concentration of the metals in contaminated water. Our study, on the other hand, provides evidence that the interactions between TiO_2_-NPs and cadmium, in addition to reducing the free concentration of individual molecules, lead to the formation of a complex that could be no longer capable to affect the health of aquatic organisms. This particular compound probably becomes unable to reach the nucleus. Based on these results, we hypothesize that TiO_2_-NPs could adsorb cadmium in contaminated aquatic environments.

Although we are far from affirming that the coexistence of engineered nanoparticles and heavy metals can be considered a favorable situation for living organisms, we showed that TiO_2_NPs modulate Cd genotoxicity in aquatic environment in vivo. These results suggest that further research is necessary to elucidate the effects induced by mixtures of NPs and heavy metals in others organisms and the type of interaction they undergo.

Recently, an interesting methodology that involved fluorescent nanoparticles for a deeper understanding of biology and medicine at the molecular level has been developed [57]. This technique is in progress in our laboratories and could help us to corroborate the hypothesis that the complex between TiO_2_- NPs and Cd that is formed can no longer penetrate the cells.

## Figures and Tables

**Figure 1 cells-10-00310-f001:**
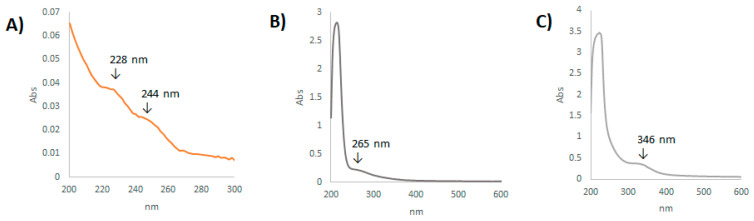
(**A**) Representative Cd UV-Vis spectrum recorded in distilled water. The spectral region 200–300 nm was highlighted; (**B**) Representative Cd UV-Vis spectrum acquired in tank water; (**C**) Cd + TiO_2_-NPs UV-Vis spectrum recorded in the range 200–600 nm.

**Figure 2 cells-10-00310-f002:**
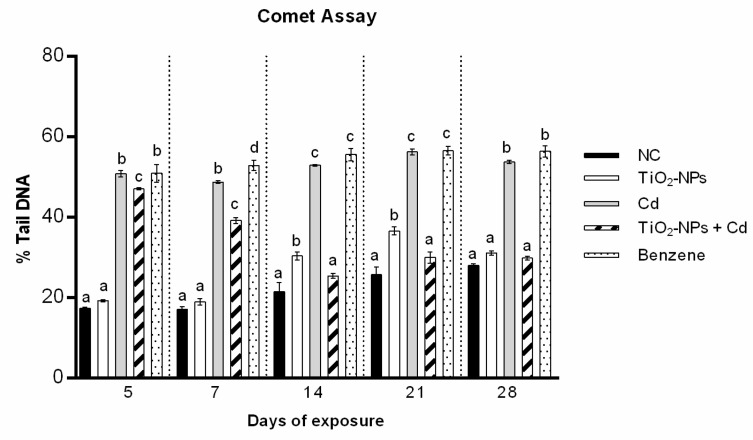
Percentage of DNA in the comet tail (ordinate) in zebrafish blood cells (*n* = 150) after different exposure times (abscissa) to TiO_2_-NPs, Cd and to TiO_2_-NPs + Cd co-exposure. The black bars are negative controls (NC); the white bars are 10 μg/L TiO_2_-NPs; the grey bars are 1 mg/L Cd; the striped bars are 10 μg/L TiO_2_-NPs + 1 mg/L Cd co-treated zebrafish. The dotted bars are positive controls (benzene 10 μL/mL). Different letters correspond to diverse statistical significances (ANOVA)* *p* ≤ 0.05.

**Figure 3 cells-10-00310-f003:**
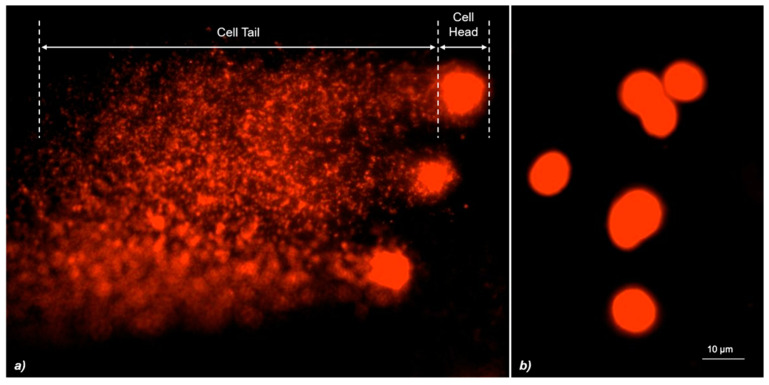
(**a**) Comet tail DNA in zebrafish blood cells analyzed using Komet software. (**b**) Undamaged zebrafish blood cells analyzed using Komet software.

**Figure 4 cells-10-00310-f004:**
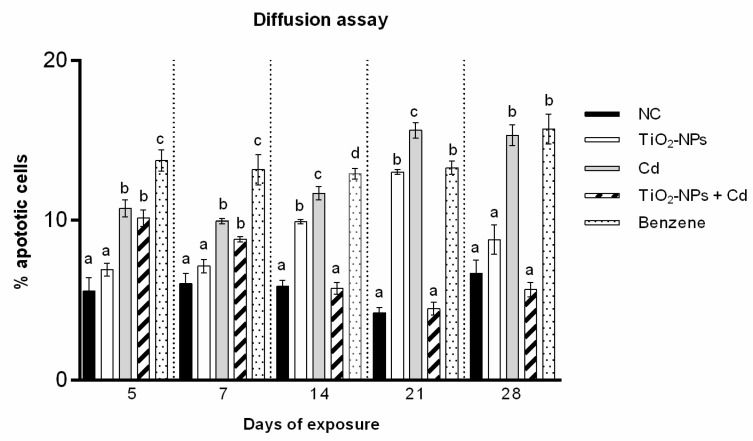
Percentage of apoptotic zebrafish blood cells (*n* = 150) (ordinate) after different exposure times (abscissa) to TiO_2_-NPs, Cd and to TiO_2_-NPs + Cd co-exposure. The black bars are negative controls (NC); the white bars are 10 μg/L TiO_2_-NPs; the grey bars are 1 mg/L Cd; the striped bars are 10 μg/L TiO_2_-NPs + 1 mg/L Cd co-treated zebrafish. The dotted bars are positive controls (benzene 10 μL/mL). Different letters correspond to diverse statistical significances (ANOVA)* *p* ≤ 0.05.

**Figure 5 cells-10-00310-f005:**
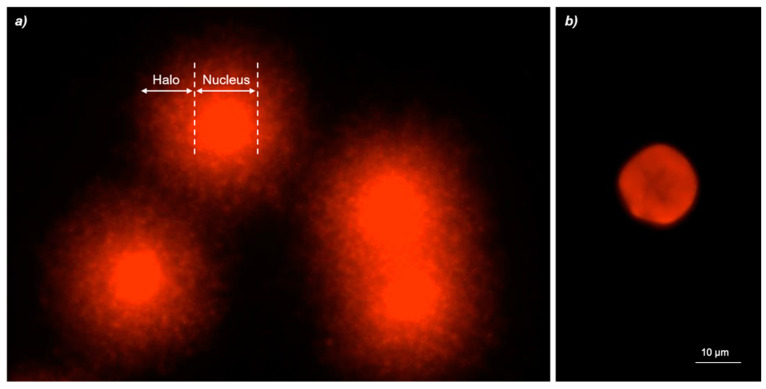
(**a**) Apoptotic zebrafish blood cells analyzed with Nikon Eclipse E-600 microscope. (**b**) Negative control zebrafish blood cell analyzed with Nikon Eclipse E-600 microscope.

**Figure 6 cells-10-00310-f006:**
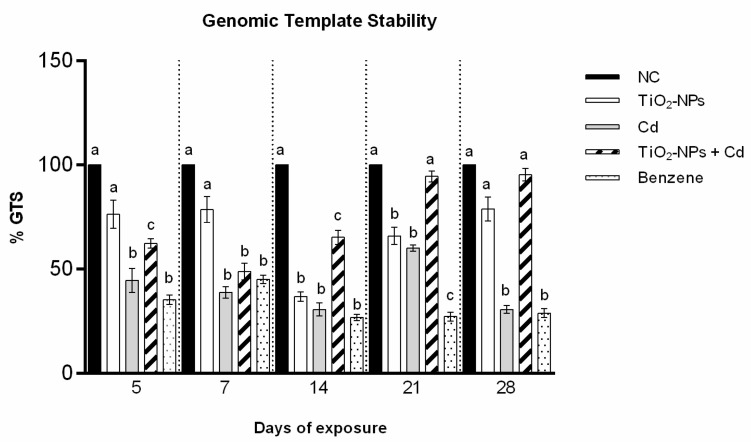
Changes in percentage of genome template stability (ordinate) in zebrafish blood cells DNA (three DNA pools for each treatment) after different exposure times (abscissa) to TiO_2_-NPs, Cd and to TiO_2_-NPs + Cd co-exposure. The black bars are negative controls (NC); the white bars are 10 μg/L TiO_2_-NPs; the grey bars are 1 mg/L Cd; the striped bars are 10 μg/L TiO_2_-NPs + 1 mg/L Cd co-treated zebrafish. The dotted bars are positive controls (benzene 10 μL/mL). Different letters correspond to diverse statistical significances (ANOVA)* *p* ≤ 0.05.

**Table 1 cells-10-00310-t001:** Molecular sizes (bp) of appeared and disappeared bands by RAPD-PCR technique in zebrafish DNA exposed to Cd (1 mg/L), to TiO_2_-NPs (10 μg/L), to Cd (1 mg/L) + TiO_2_-NPs (10 μg/L) and to benzene (10 µL/mL) for different exposure times. ^a^ Control bands are at: 200, 220, 250, 290, 320, 400, 450, 500, 550, 600, 800 bp.

Treatment	Exposure Days	Gained Bands	Lost Bands ^a^
Cd (1 mg/L)	5 7 14 21 28	650, 1000 690 350, 650, 690, 780, 850 - 180, 300, 420, 850	200, 220, 290, 320 200, 230, 290, 320, 550, 800 290, 320 220, 250, 320, 600 290, 400, 600, 800
TiO_2_-NPs (10 μg/L)	5 7 14 21 28	100, 690 100, 690 100, 420, 690 100, 420 380, 420, 480	220 220 220, 250, 300, 320 320, 500, 550 -
TiO_2_-NPs (10 μg/L) + Cd (1 mg/L)	5 7 14 21 28	100, 420, 480, 690 300, 420, 480 180, 850 - -	- 200, 220 290, 320 550 600
Benzene (10 µL/mL)	5 7 14 21 28	180, 650, 690 - 100, 580, 650, 690 150, 180, 580, 690 150, 420, 430, 850	200, 220, 290, 320 200, 230, 290, 320, 550, 800 290, 320, 450, 550 250, 400, 600, 800 200, 400, 550, 800

## Data Availability

Not applicable.

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
