# Peer review of "Adsorption of Cd to TiO2-NPs Forms Low Genotoxic Aggregates in Zebrafish Cells"

_cells, 2021, doi:10.3390/cells10020310_

Round 1

Reviewer 1 Report

The authors (Rocco and team) have previously shown that TiO2 is toxic to zebrafish (as a model for aquatic toxicology). It has not been determined if co-exposure of TiO2 with other metals is as toxic to zebrafish, or protective, so here Mottola et al compare exposure to TiO2 on its own with exposure to TiO2 plus cadmium. Overall, they show that when TiO2 and Cd are given in combination, there is less toxicity to zebrafish than if each compound is given separately. However, there is no evidence to support the conclusion that there is less toxicity because one metal aggregates with the other as suggested in the title "Adsorption of Cd to TiO2-NPs forms low genotoxic aggregates in zebrafish cells", or in the abstract, and discussion. Furthermore, not enough methodological information was provided to determine if the data are correct.

Methods:

  1. Clarification is needed on the experimental groups. It is stated that experiments were repeated in triplicate, and that the animal group size was 20, but that 132 zebrafish were used. Both cannot be correct. If there is 20 fish per group, and experiments performed in triplicate, then:
    TiO2 alone = 3x20 = 60
    TiO2 + Cd = 3x20 = 60
    -ve control = 3x20 = 60
    +ve control = 3x20 = 60
    Then, 5 time points. How many fish per time point?
    Therefore 132 fish are too few for the experiment. Please describe the numbers of fish used for the experiment and for each time point accurately. On top of the number of individuals, the number of cells/nuclei analysed per individual should be detailed (see further comments below).
  2. How frequently was the medium/tank water changed during the long exposures? Was the water replaced regularly?
  3. Taking of blood for the experiments - the methodology here is unclear. Is blood sampling a survivable procedure in this study? Or were animals sacrificed? How was blood obtained from very young animals (5-14 days)? This would be extremely difficult, and also unlikely to be survivable.
  4. How many fish died during the treatments, during long exposure? It's not clear if individuals are sacrificed for each experiment or the same individual is sampled more than once, the methods do not make this clear. 
  5. Statistics - it is stated that a student's t-test was used to determine statistical significance. Because more than two conditions are being compared, the authors should instead use ANOVA.

Results:

Figures 2 and 3 do not contain enough information to determine the veracity of the results they describe. The robustness of the data would be improved by including some primary results and a better description of numbers.

Please include representative images of the comet and apoptotic cell assays. Show how % tail DNA was calculated for comets. There can be a lot of variability in comet assays, how was it determined which nuclei to include?

  1. How many nuclei/cells does each bar in the graphs represent?
  2. How many zebrafish individuals does each bar in the graphs represent?
  3. How was the experiment replicated?

Table 1, summarising the RAPD results. Interpretation of the data would be assisted if the authors would show the corresponding gels. It is not clear what the changes in band identity signifies in biological terms, because PCRs can be affected by sample quality, PCR efficiency, etc. Intactness of DNA can reflect the isolation and storage of samples rather than in situ DNA damage, so how is this controlled for? How was this experiment replicated?

Figure 4
1. The authors need to clearly demonstrate how they got these results.
2. Again, it needs to be stated how many individuals the bars represent, what is the replication, and the stats should be ANOVA.

Discussion:

Regarding the statement "Our characterization results showed that the two substances aggregate in an aquatic environment as Cd adsorb on TiO2-NPs and probably form a compound unable to penerate the cells and damage the DNA." While this might be a reasonable assumption, the data presented do not directly support this conclusion. Can the authors prove that a complex between TiO2 and Cd is formed and that this can no longer penetrate cells, eg by using fluorescent NPs? 

Author Response

We are very grateful to you for the careful reading and revision that were made in order to improve our manuscript. We appreciate your time and the detailed valuable comments, which improved the quality of the manuscript. Answers are given below, point by point.

Comments and Suggestions for Authors

The authors (Rocco and team) have previously shown that TiO2 is toxic to zebrafish (as a model for aquatic toxicology). It has not been determined if co-exposure of TiO2 with other metals is as toxic to zebrafish, or protective, so here Mottola et al compare exposure to TiO2 on its own with exposure to TiO2 plus cadmium. Overall, they show that when TiO2 and Cd are given in combination, there is less toxicity to zebrafish than if each compound is given separately. However, there is no evidence to support the conclusion that there is less toxicity because one metal aggregates with the other as suggested in the title "Adsorption of Cd to TiO2-NPs forms low genotoxic aggregates in zebrafish cells", or in the abstract, and discussion. Furthermore, not enough methodological information was provided to determine if the data are correct.

Methods:

  • Clarification is needed on the experimental groups. It is stated that experiments were repeated in triplicate, and that the animal group size was 20, but that 132 zebrafish were used. Both cannot be correct. If there is 20 fish per group, and experiments performed in triplicate, then:

TiO2 alone = 3x20 = 60

TiO2 + Cd = 3x20 = 60

-ve control = 3x20 = 60

+ve control = 3x20 = 60

Then, 5 time points. How many fish per time point?

Therefore 132 fish are too few for the experiment. Please describe the numbers of fish used for the experiment and for each time point accurately. On top of the number of individuals, the number of cells/nuclei analysed per individual should be detailed (see further comments below).

R: thank you for your important observation. There was an inaccuracy in the experimental design. Experiments were not repeated in triplicate, while the assays were performed in triplicate. Therefore, we deleted the sentence "experiments were performed in triplicate" in paragraph 2.2, and added "the assay was performed in triplicate" in paragraph 2.5, 2.6, 2.7. We used 150 zebrafish and each group was constituted by 30 individuals. In detail 6 fish for each time point (5, 7, 14, 21, 28 days). We have corrected in the text at 2.3 section (Specimens preparation).

  • How frequently was the medium/tank water changed during the long exposures? Was the water replaced regularly?

R: The water was replaced and the substances were dissolved at the chosen concentration every 7 days. We added this detail in 2.3 paragraph.

  • Taking of blood for the experiments - the methodology here is unclear. Is blood sampling a survivable procedure in this study? Or were animals sacrificed? How was blood obtained from very young animals (5-14 days)? This would be extremely difficult, and also unlikely to be survivable. How many fish died during the treatments, during long exposure? It's not clear if individuals are sacrificed for each experiment or the same individual is sampled more than once, the methods do not make this clear.

R: Regarding these important observations, we specify that we used adult animals as already reported in paragraph 2.2. The numbers 5,7,14,21 and 28 refer to the exposure days and not to the age of specimens. The fish did not die during the treatments. The animals were sacrificed to collect blood and muscle for each experiment. We used the anaesthetic according to the Guide for Use and Care of Laboratory Animals (European Communities Council Directive). We added this in paragraph 2.3.

  • Statistics - it is stated that a student's t-test was used to determine statistical significance. Because more than two conditions are being compared, the authors should instead use ANOVA.

R: Thank you for your suggestion. We used the unpaired Student’s t-test to compare just two groups at a time. The mean results for each treatment with respect to the mean results for the negative control.

Results:

  • Figures 2 and 3 do not contain enough information to determine the veracity of the results they describe. The robustness of the data would be improved by including some primary results and a better description of numbers.

R: Thank you for your observations. We have completely rewritten paragraphs 3.2 and 3.3.

  • Please include representative images of the comet and apoptotic cell assays. Show how % tail DNA was calculated for comets. There can be a lot of variability in comet assays, how was it determined which nuclei to include?

R: Thank you for this advice. Representative images of the comet and apoptotic were added (Figure 2b and Figure 3b). We calculated the % tail DNA using the Komet software ver.6.0.0: we have specified in paragraph 2.5. In detail, the software considers the percentage of DNA in the tail as an indicator of DNA damage, and is able to calculate and produce this important measure of damage for each cell analyzed. DNA fragments migrate away from the “head” into the “tail” based on their size, and the intensity of the comet tail relative to the total intensity (head plus tail) reflects the amount of DNA breakage. We randomly chose 50 nuclei to be analyzed for each slide and then performed a statistical analysis of the results vs the controls.

  • How many nuclei/cells does each bar in the graphs represent?

R: Each bar in the graphs represent 150 zebrafish blood cells.

  • How many zebrafish individuals does each bar in the graphs represent?

R: None. The tests were performed on zebrafish blood cells.

  • How was the experiment replicated?

R: Three slides for sample were used for each experiment and 50 cells were analysed for each slide. We added this detail in paragraphs 2.5 and 2.6.

  • Table 1, summarising the RAPD results. Interpretation of the data would be assisted if the authors would show the corresponding gels. It is not clear what the changes in band identity signifies in biological terms, because PCRs can be affected by sample quality, PCR efficiency, etc.

R: Thanks for this important comment. The changes in band are linked to molecular events. In particular, the appearance of bands identify that DNA damage can be due to point mutations and/or DNA rearrangements; while the disappearance of bands could be due to the formation of DNA adducts or double strand breaks also (Liu et al., 2007). We added this explanation and reference nr.37 in the text at 2.7 paragraph.

  •  Intactness of DNA can reflect the isolation and storage of samples rather than in situ DNA damage, so how is this controlled for?

R: The DNA purity was evaluated by Nanodrop 2000 (Thermo Scientific) while DNA integrity thanks to electrophoresis on 2% agarose gel and the samples were analysed by RAPD-PCR immediately after DNA isolation. We added this information in 2.7 section.

  •  How was this experiment replicated?

R: We isolated DNA from each treated zebrafish and made three DNA pools for each treatment. Then each DNA pool were undergoing to RAPD amplification and electrophoresis in triplicate. We added this detail in paragraphs 2.7.

Figure 4

  • The authors need to clearly demonstrate how they got these results.

R: As indicated in paragraph 2.7, the %,GTS, shown in figure 4, was calculated as follows: %,GTS= (1-a/n) x 100, where a is the average number of polymorphic bands detected in each exposed sample and n the number of total bands in the non-treated cells (electrophoretic profile of negative control). We have added the reference nr. 36 in paragraph 2.7 to clarify the % GTS calculation: “Rocco L, Valentino IV, Scapigliati G, Stingo V (2014) RAPD-PCR analysis for molecular characterization and genotoxic studies of a new marine fish cell line derived from Dicentrarchus labrax. Cytotechnology, 66(3):383-93”. In details, polymorphic bands include bands that disappeared compared to the bands detected in the negative control and the bands that de novo appeared (not present in the negative control) in treated sample. Therefore, by counting the number of polymorphic bands in each sample with respect to the bands present in the negative control, the genomic stability of the template is obtained.

  • Again, it needs to be stated how many individuals the bars represent, what is the replication, and the stats should be ANOVA.

R: In figure 4 the bars do not represent the treated fish but indicate the average of the percentage of Genomic Template Stability for each experimental group. We used the unpaired Student’s t-test to compare just two groups: the means results of each treatment with respect to the means results of the negative control.

Discussion:

  • Regarding the statement "Our characterization results showed that the two substances aggregate in an aquatic environment as Cd adsorb on TiO2-NPs and probably form a compound unable to penerate the cells and damage the DNA." While this might be a reasonable assumption, the data presented do not directly support this conclusion. Can the authors prove that a complex between TiO2 and Cd is formed and that this can no longer penetrate cells, eg by using fluorescent NPs?

R: Thanks to the results from the characterization reported in paragraph 3.1, we have evidenced that the two substances aggregate in an aquatic environment. We appreciate this important advice, so we will evaluate if the complex between TiO2-NPs and Cd that is formed can no longer penetrate the cells using fluorescent NPs in a future study. This technique is in progress in our laboratories. We have added this future perspective at the end of the discussion.

Reviewer 2 Report

The article presents an interesting problem about the possible role of nanoparticles in bioremediation processes of heavy metal contamination.

However, all the methods tested show indirect results and this leads to somewhat speculative conclusions.

Comet and Diffusion Assays are performed on blood cells, while RAPD-PCR studies use muscle cells. The authors should clarify why they use different tissues in the study, and how extrapolated the results are, using such dissimilar tissues.

The experimental design of the Cometa Assay studies are unclear, and with the data provided, it would be impossible to repeat these assays for corroboration. Authors should clearly explain how these quantifications were performed. Although the diffusion assay is accepted primarily for qualitative studies, it is ill-suited for quantitative studies. Again, in the M&M the calculation strategy must be detailed and also how the statistics of these analyzes have been carried out.

On the other hand, the results of the RAPD-PCR technique are confusing and the authors should endeavor to summarize the main findings rather than repeat what is indicated in Table 1 (which is redundant with the text).

Although the English language is moderately correct, it would be desirable for the article to be revised by a native speaker to correct some grammatical defects.

Author Response

We are very grateful to you for the careful reading and revision that were made in order to improve our manuscript. We appreciate your time and the detailed valuable comments, which improved the quality of the manuscript. Answers are given below, point by point.

Comments and Suggestions for Authors

The article presents an interesting problem about the possible role of nanoparticles in bioremediation processes of heavy metal contamination.

  • However, all the methods tested show indirect results and this leads to somewhat speculative conclusions.

R: Thanks for this interesting observation. Although our methods regarding the genotoxicity of TiO2-NPs and Cd evaluation evidenced indirect results, the characterization and analytical determinations results show the formation of a stable complex between the two substances. Therefore, we could hypothesize that the reduction of genotoxicity is probably due to the formation of TiO2-NPs + Cd aggregates in aquatic environment and consequently the bioremediation represents only a hypothesis. We have clarified this deduction in the discussion section as follow: “Our study, on the other hand, provides evidence that the interactions between TiO2-NPs and cadmium, in addition to reducing the free concentration of individual molecules, lead to the formation of a complex that could be no longer capable to affect the health of aquatic organisms. This particular compound probably becomes unable to reach the nucleus. Based on these results, we hypothesize that TiO2-NPs could adsorb cadmium in contaminated aquatic environments”. Moreover, we deleted the last sentence with speculative conclusion: “in order to establish whether TiO2-NPs nanoparticles can be used in a controlled manner as bio-remedies for aquatic contaminants”.

  • Comet and Diffusion Assays are performed on blood cells, while RAPD-PCR studies use muscle cells. The authors should clarify why they use different tissues in the study, and how extrapolated the results are, using such dissimilar tissues.

R: Thanks for this important comment. We have clarified the use of different tissue in discussion section as follow: As TiO2-NPs could facilitate the transport of some contaminants into internal tissues [48], we have evaluated the genotoxicity in different tissues (muscle and erythrocytes) in order to fully investigate the genotoxicity given by the co-exposure of TiO2-NPs and Cd, as previously performed in the marine environment [3]. We have added reference nr. 48: “B.Pan, B.S. Xing, "Adsorption mechanisms of organic chemicals on carbon nanotubes", Environ. Sci. Technol., vol. 42, pp. 9005-9013, Nov. 2008” in the discussion.

  • The experimental design of the Comet Assay studies are unclear, and with the data provided, it would be impossible to repeat these assays for corroboration. Authors should clearly explain how these quantifications were performed.

R: Thanks for this important observation. We have explained how we quantified the damage by Comet assay in paragraph 2.5 as follow: Comets were selected without bias and represented the entire gel as recommended by Collins AR (2004). The images from the comet slides were analysed by the software by drawing a measurement frame on the screen around the comet area. The selected measurement parameters were saved and the statistical analysis was conducted. We considered relative fluorescence intensity of head and tail parameter (normally expressed as a percentage of DNA in tail). The assay was performed in triplicate, three slides for sample were used for each experiment and 50 cells were analyzed for each slide. We have added reference nr. 31: “R. Collins, "The comet assay for DNA damage and repair: principles, applications, and limitations," Mol Biotechnol., vol. 26(3), pp. 249-61, Mar 2004” in paragraph 2.5.

  • Although the diffusion assay is accepted primarily for qualitative studies, it is ill-suited for quantitative studies. Again, in the M&M the calculation strategy must be detailed and also how the statistics of these analyzes have been carried out.

R: The Diffusion assay is a simple, sensitive, and reliable assay for the quantification of apoptosis where the number of cells with apoptotic appearance can be scored and quantified by using a fluorescent microscope according to Singh, 2000 [31]. We have added in paragraph 2.6 how we quantified apoptotic cells and the statistical analysis carried out: “We observed the slides using an epifluorescence microscope (Nikon Eclipse E-600) with a BP 515-560 nm filter and an LP 580 nm filter. The Diffusion assay slides were scored by subdividing the degree of DNA diffusion pattern in five damage classes as reported by Cantafora and collaborators [33]. In details, we considered only class 5 (apoptotic cell). The number of apoptotic cells were quantified as the percentage of cells with apoptotic appearance on total cells by using a fluorescent microscope. The assay was performed in triplicate: three slides for sample were used for each experiment and 50 cells were analysed for each slide.”

  • On the other hand, the results of the RAPD-PCR technique are confusing and the authors should endeavor to summarize the main findings rather than repeat what is indicated in Table 1 (which is redundant with the text).

R: Thanks for this important comment. We rewrote the paragraph 2.7 summarizing the results of RAPD-PCR technique as follows: The electrophoretic profiles of each treated samples showed a different pattern from the negative control after each exposure days. The exposure to Cd induced six bands variation after 5 days, seven bands variations after 7 and 14 days, while four and eight bands variations after 21 and 28 days respectively. TiO2-NPs induced a variation of three bands after 5, 7 and 28 exposure days while seven and five bands after 14 and 21 exposure days respectively. The electrophoretic profiles obtained after Cd + TiO2-NPs exposure showed four bands changes after 5 and 14 exposure days; five bands changes after 7 days and one band change after 21 and 28 exposure days respect to negative control.

  • Although the English language is moderately correct, it would be desirable for the article to be revised by a native speaker to correct some grammatical defects.

R: We carried out a review of the English language by a native speaker colleague

Round 2

Reviewer 1 Report

The authors have clarified most of my questions and greatly improved the manuscript. It's much easier now to understand how the experiment was done. Please give the age of the zebrafish at the onset of the experiment for further clarity.

The inclusion of images in figures 2 and 3 is appreciated, but these need to be annotated because it is difficult to determine what one is looking at. For example, please indicate the features of comets/apoptotic cells relative to control/undamaged cells. The comet image can be used to indicate which portion was used to calculate % DNA in tail.

To help the reader understand what the data represent, it would be helpful to include in the legend what the bars in the bar graphs represent, eg, if the bars represent 150 zebrafish blood cells from pools of x number of zebrafish, please say that in the legend.

I am still not sure if the t-test 'treatment vs control' is the best statistical formula to use because the authors want to compare TiO2 and Cd combined, with TiO2 or Cd on their own, relative to the controls. The statistics should represent whether there is a significant difference between TiO2+Cd, TiO2 alone, Cd alone.

Author Response

Thank you very much for your nice and encouraging comments. We have addressed all of your comments and revised the manuscript accordingly. All the revised parts are tracked throughout the manuscript. A review of the English language by a native speaker colleague was carried out. Answers are given below, point by point.

The authors have clarified most of my questions and greatly improved the manuscript. It's much easier now to understand how the experiment was done. Please give the age of the zebrafish at the onset of the experiment for further clarity.

R: Thank you. We have added the age of the zebrafish at paragraph 2.2 “Specimens preparation” as follow: “Experiments were conducted on 150 adult 8-month-old zebrafish purchased from a local supplier”.

The inclusion of images in figures 2 and 3 is appreciated, but these need to be annotated because it is difficult to determine what one is looking at. For example, please indicate the features of comets/apoptotic cells relative to control/undamaged cells. The comet image can be used to indicate which portion was used to calculate % DNA in tail.

R: Thank you for your suggestions. We have added new figures. In detail, Figure 3 shows comet cells compared to undamaged cells, while Figure 5 exhibits apoptotic cells relative to control. For each figure we indicated the portions used to calculate DNA damage.

To help the reader understand what the data represent, it would be helpful to include in the legend what the bars in the bar graphs represent, eg, if the bars represent 150 zebrafish blood cells from pools of x number of zebrafish, please say that in the legend.

R: Thank you for this comment. We have added the number of zebrafish blood cells (n=150) in Figure 2 and 4 legend. In figure 6 legend, we have added the number of DNA pool: three DNA pools for each treatment.

I am still not sure if the t-test 'treatment vs control' is the best statistical formula to use because the authors want to compare TiO2 and Cd combined, with TiO2 or Cd on their own, relative to the controls. The statistics should represent whether there is a significant difference between TiO2+Cd, TiO2 alone, Cd alone.

R: We thank you for your additional observation. In this regard, we analysed the data by ANOVA (ANalysis Of VAriances) test, indicating the significant differences for each treatment. We added the ANOVA test in paragraph 2.8 "Statistical analysis" and on figures 2,4,6.